# NNGeometry: Easy and Fast Fisher Information Matrices and Neural Tangent Kernels in PyTorch

## Abstract

Fisher Information Matrices (FIM) and (finite-width) Neural Tangent Kernels (NTK) are useful tools in a number of diverse applications related to neural networks Pascanu & Bengio (2013); Kirkpatrick et al. (2017); Wu et al. (2017); Liang et al. (2019); Du et al. (2018). Yet these theoretical tools are often difficult to implement using current libraries for practical size networks, given that they require per-example gradients, and a large amount of memory since they scale as the number of parameters (for the FIM) or the number of examples $\times$ cardinality of the output space (for the NTK). NNGeometry is a PyTorch library that offers a simple interface for computing various linear algebra operations such as matrix-vector products, trace, frobenius norm, and so on, where the matrix is either the FIM or the NTK, leveraging recent advances in approximating these matrices. We hereby introduce the library and motivate our design choices, then we demonstrate it on modern deep neural networks.

Code for this paper is available at this (anonymized) repo: `https://github.com/OtUmm7ojOrv/nngeometry`.

Practical and theoretical advances in deep learning have been accelerated by the development of an ecosystem of libraries allowing practitioners to focus on developing new techniques instead of spending weeks or months re-implementing the wheel. In particular, automatic differentiation frameworks such as Theano (Bergstra et al., 2011), Tensorflow (Abadi et al., 2016) or PyTorch (Paszke et al., 2019) have been the backbone for the leap in performance of last decade's increasingly deeper neural networks as they allow to compute average gradients efficiently, used in the stochastic gradient algorithm or variants thereof. While being versatile in neural networks that can be designed by varying the type and number of their layers, they are however specialized to the very task of computing these average gradients, so more advanced techniques can be burdensome to implement.

While the popularity of neural networks has grown thanks to their always improving performance, other techniques have emerged, amongst them we highlight some involving Fisher Information Matrices (FIM) and Neural Tangent Kernels (NTK). Approximate 2nd order (Schraudolph, 2002) or natural gradient techniques (Amari, 1998) aim at accelerating training, elastic weight consolidation (Kirkpatrick et al., 2017) proposes to fight catastrophic forgetting in continual learning and WoodFisher (Singh & Alistarh, 2020) tackles the problem of network pruning so as to minimize its computational footprint while retaining prediction capability. These 3 methods all use the Fisher Information Matrix while formalizing the problem they aim at solving, but resort to using different approximations when going to implementation. Similarly, following the work of Jacot et al. (2018), a line of work study the NTK in either its limiting infinite-width regime, or during training of actual finite-size networks.

All of these papers start by formalizing the problem at hand in a very concise math formula, then face the experimental challenge that computing the FIM or NTK involves performing operations for which off-the-shelf automatic differentiation libraries are not well adapted. An even greater turnoff comes from the fact that these matrices scale with the number of parameters (for the FIM) or the number of examples in the training set (for the empirical NTK). This is prohibitively large for modern neural networks involving millions of parameters or large datasets, a problem circumvented by a series of techniques to approximate the FIM (Ollivier, 2015; Martens & Grosse, 2015; George

**using a KFAC Fisher**

```
1  F_kfac = FIM(model=model,
2              loader=loader,
3              representation=PMatKFAC,
4              n_output=10)
5
6  v = PVector.from_model(model)
7
8  vTMv = F_kfac.vTMv(v)
```

**using implicit computation**

```
1  F_full = FIM(model=model,
2              loader=loader,
3              representation=PMatDense,
4              n_output=10)
5
6  v = PVector.from_model(model)
7
8  vTMv = F_full.vTMv(v)
```

Figure 1: Computing a vector-Fisher-vector product $\mathbf{v}^\top F \mathbf{v}$, for a 10-fold classification model defined by `model`, can be implemented with the same piece of code for 2 representations of the FIM using NNGeometry, even if they involve very different computations under the hood.

et al., 2018). NNGeometry aims at making use of these approximations effortless, so as to accelerate development or analysis of new techniques, allowing to spend more time on the theory and less time in fighting development bugs. NNGeometry's interface is designed to be as close as possible to maths formulas. In summary, this paper and library contribute:

- We introduce NNGeometry by describing and motivating design choices.
  - A unified interface for all FIM and NTK operations, regardless of how these are approximated.
  - Implicit operations for ability to scale to large networks..
- Using NNGeometry, we get new empirical insights on FIMs and NTKs:
  - We compare different approximations in different scenarios.
  - We scale some NTK evolution experiments to TinyImagenet.

## 1 PRELIMINARIES

### 1.1 NETWORK LINEARIZATION

Neural networks are parametric functions $f(x, \mathbf{w}) : \mathcal{X} \times \mathbb{R}^d \to \mathbb{R}^c$ where $x \in \mathcal{X}$ are covariates from an input space, and $\mathbf{w} \in \mathbb{R}^d$ are the network's parameters, arranged in layers composed of weight matrices and biases. The function returns a value in $\mathbb{R}^c$, such as the $c$ scores in softmax classification, or $c$ real values in $c$-dimensional regression. Neural networks are trained by iteratively adjusting their parameters $\mathbf{w}^{(t+1)} \leftarrow \mathbf{w}^{(t)} + \delta \mathbf{w}^{(t)}$ using steps $\delta \mathbf{w}^{(t)}$ typically computed using the stochastic gradient algorithm or variants thereof, in order to minimize the empirical risk of a loss function.

In machine learning, understanding and being able to control the properties of the solution obtained by an algorithm is of crucial interest, as it can provide generalization guarantees, or help design more efficient or accurate algorithms. Contrary to (kernelized) linear models, where closed-form expressions of the empirical risk minimizer exist, deep networks are non-linear functions, whose generalization properties and learning dynamics is not yet fully understood. Amongst the recent advances toward improving theory, is the study of the linearization (in $\mathbf{w}$) of the deep network function $f(x, \mathbf{w})$:

$$f(x, \mathbf{w} + \delta \mathbf{w}) = f(x, \mathbf{w}) + J(x, \mathbf{w}) \delta \mathbf{w} + o(\|\delta \mathbf{w}\|) \,^1 \tag{1}$$

where $J(x, \mathbf{w}) = \frac{\partial f(x, \mathbf{w})}{\partial \mathbf{w}}$ is the Jacobian with respect to parameters $\mathbf{w}$, computed in $(\mathbf{w}, x)$, mapping changes in parameter space $\delta \mathbf{w}$ to corresponding changes in output space using the identity $\delta f(x, \mathbf{w}, \delta \mathbf{w}) = J(x, \mathbf{w}) \delta \mathbf{w}$. For tiny steps $\delta \mathbf{w}$, we neglect the term $o(\|\delta \mathbf{w}\|)$ thus $f$ is close to its linearization. It happens for instance at small step sizes, or in the large-width limit with the specific parameter initialization scheme proposed by Jacot et al. (2018).

---

[1]The Landau notation $o$ (pronounced "little-o") means a function whose exact value is irrelevant, with the property that $\lim_{x \to 0} \frac{o(x)}{x} = 0$, or in other words that is negligible compared to $x$ for small $x$.

## 1.2 PARAMETER SPACE METRICS AND FISHER INFORMATION MATRIX

While neural networks are trained by tuning their parameters $\mathbf{w}$, the end goal of machine learning is not to find the best parameter values, but rather to find *good* functions, in a sense that is dependent of the task at hand. For instance different parameter values can represent the same function (Dinh et al., 2017). On the contrary 2 parameter space steps $\delta\mathbf{w}_1$ and $\delta\mathbf{w}_2$ with same euclidean norm can provide very different changes in a function ($\delta f(x, \mathbf{w}, \delta\mathbf{w}_1) \neq \delta f(x, \mathbf{w}, \delta\mathbf{w}_2)$). In order to quantify changes of a function, one generally defines a distance[2] on the function space. Examples of such distances are the $L_k$-norms, Wasserstein distances, or the KL divergence used in information geometry.

To each of these function space distances correspond a parameter space metric. We continue our exposition by focusing on the KL divergence, which is closely related to the Fisher Information Matrix, but our library can be used for other function space distances. Suppose $f$ is interpreted as log-probability of a density $p$: $\log p(x, \mathbf{w}) = f(x, \mathbf{w})$, the KL divergence gives a sense of how much the probability distribution changes when adding a small increment $\delta\mathbf{w}$ to the parameters of $f(x, \mathbf{w})$. We can approximate it as:

$$\text{KL}\left(p(x, \mathbf{w}) \, \| \, p(x, \mathbf{w} + \delta\mathbf{w})\right) = \int_{x \in \mathcal{X}} \log\left(\frac{p(x, \mathbf{w})}{p(x, \mathbf{w} + \delta\mathbf{w})}\right) dp(x, \mathbf{w}) \tag{2}$$

$$= \frac{1}{2} \int_{x \in \mathcal{X}} \left(\frac{1}{p(x, \mathbf{w})} J(x, \mathbf{w}) \delta\mathbf{w}\right)^2 dp(x, \mathbf{w}) + o\left(\|\delta\mathbf{w}\|^2\right) \tag{3}$$

where we used this form (derived in appendix) in order to emphasize how steps in parameter space $\delta\mathbf{w}$ affect distances measured on the function space: equation 3 is the result of i) taking a step $\delta\mathbf{w}$ in parameter space; ii) multiplying with $J(x, \mathbf{w})$ to push the change to the function space; iii) weight this function space change using $p(x, \mathbf{w})^{-1}$; iv) square and sum. In particular, because of the properties of the KL divergence, there is no second derivative of $f$ involved, even if equation 3 is equivalent to taking the 2nd order Taylor series expansion of the KL divergence. We can rewrite in a more concise way:

$$\text{KL}\left(f(x, \mathbf{w}) \, \| \, f(x, \mathbf{w} + \delta\mathbf{w})\right) = \delta\mathbf{w}^\top F_{\mathbf{w}} \delta\mathbf{w} + o\left(\|\delta\mathbf{w}\|^2\right) \tag{4}$$

which uses the $d \times d$ FIM $F_{\mathbf{w}} = \int_{x \in \mathcal{X}} \frac{1}{p(x, \mathbf{w})^2} J(x, \mathbf{w})^\top J(x, \mathbf{w}) \, dp(x, \mathbf{w})$. In particular, we can now define the norm $\|\delta\mathbf{w}\|_{F_{\mathbf{w}}} = \delta\mathbf{w}^\top F_{\mathbf{w}} \delta\mathbf{w}$ used in the natural gradient algorithm (Amari (1998), also see Martens (2020) for a more thorough discussion of the FIM), in elastic weight consolidation (Kirkpatrick et al., 2017), or in pruning (Singh & Alistarh, 2020). Other quantities also share the same structure of a covariance of parameter space vectors, such as the covariance of loss gradients in TONGA (Roux et al., 2008), the second moment of loss gradients[3] (Kunstner et al., 2019; Thomas et al., 2020), or posterior covariances in bayesian deep learning (e.g. in Maddox et al. (2019)).

## 1.3 NEURAL TANGENT KERNEL

Another very active line of research around the linearization of equation 1 is to take inspiration from the rich literature on kernel methods by defining the neural tangent kernel (NTK):

$$k_{\mathbf{w}}(x, y) = J(x, \mathbf{w}) J(y, \mathbf{w})^\top \tag{5}$$

In the limit of networks infinite width, Jacot et al. (2018) have shown that the tangent kernel remains constant through training using gradient descent, which allows to directly apply kernel learning theory to deep learning. While this regime is of theoretical interest, it arguably does not explain what happens at finite width, where the NTK evolves during training.

While kernels are functions of the whole input space $\mathcal{X} \times \mathcal{X}$, we often only have access to a limited number of samples in a datasets. We thus resort to using the kernel evaluated at points $x_i$ of a

---

[2]We here use the notion of distance informally.

[3]The second moment of loss gradients is sometimes called *empirical Fisher*.

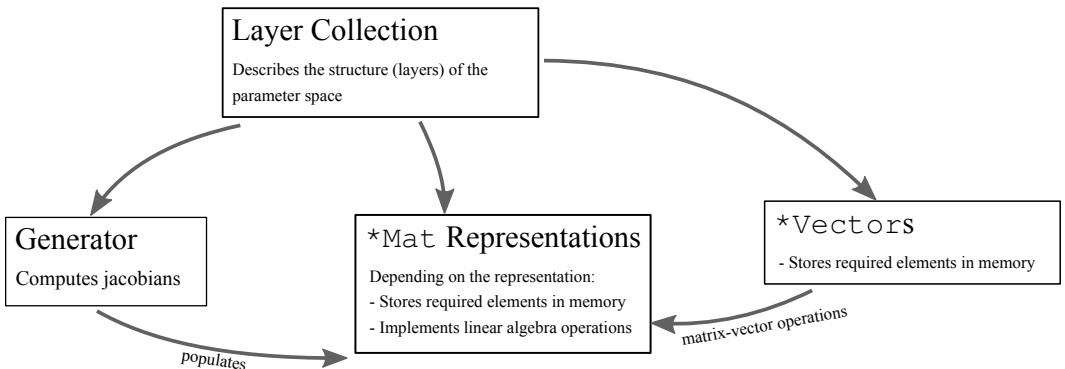

Figure 2: Schematic description of NNGeometry's main components

training or a test set, called the Gram Matrix $(K_{\mathbf{w}})_{ij} = k_{\mathbf{w}}(x_i, x_j)$. Note that in the case where the output space is multidimensional with dimension $c$, then $K_{\mathbf{w}}$ is in fact a 4d tensor.

## 2 DESIGN AND IMPLEMENTATION

### 2.1 DIFFICULTIES

Current deep learning frameworks such as PyTorch and Tensorflow are well adapted to neural network training, i.e. computing average gradients over parameters, used in optimizers such as Adam and others. However, when going to more advanced algorithms or analysis techniques involving FIMs and NTKs, practitioners typically have to hack the framework's internal mechanisms, which is time consuming, error prone, and results in each project having its own slightly different implementation of the very same technique. We here list the difficulties in computing FIMs and NTKs using current frameworks:

**Per-example gradient** FIMs and NTKs require per-example Jacobians $J(x_i, \mathbf{w})$ of a dataset $(x_i)_i$. This can be obtained by looping through examples $x$, but at the cost of not using mini-batched operations, thus missing the benefit of using GPUs. NNGeometry's Jacobian generator extensively use efficient techniques such as Goodfellow (2015)

**Memory usage and computational cost** A FIM matrix is $d \times d$ where $d$ is the total number of parameters. With a memory cost in $\mathcal{O}(d^2)$, this is prohibitively costly even for moderate size networks. Typical linear algebra operations have a computational cost in either $\mathcal{O}(d^2)$ (e.g. matrix-vector product) or even $\mathcal{O}(d^3)$ (e.g. matrix inverse). NNGeometry instead comes with recent lower memory intensive approximations.

### 2.2 NNGEOMETRY'S DESIGN

#### 2.2.1 ABSTRACT OBJECTS

In section 1, we have worked with abstract mathematical objects $\delta \mathbf{w}$, $\delta f(x, \mathbf{w}, \delta \mathbf{w})$, $J(x, \mathbf{w})$, $F_{\mathbf{w}}$ and $K_{\mathbf{w}}$. We now identify these mathematical objects to Python classes in NNGeometry.

We start with the parameter space, that we previously identified as $\mathbb{R}^d$. Closer to how they are actually implemented in deep learning frameworks, vectors in the parameter space $\mathbf{w}$ can equivalently be considered as a set of weight matrices and bias vectors $\mathbf{w} = \{W_1, b_1, \ldots, W_l, b_l\}$. Parameter space vectors are represented by the class `PVector` in NNGeometry, which is essentially a dictionary of PyTorch `Parameters`, with basic algebra logic: `PVectors` can be readily added, substracted, and scaled by a scalar with standard python operators. As an illustration `wsum = w1 + w2` internally loops through all parameter tensors of `w1` and `w2` and returns a new `PVector w_sum`.

Similarly, and more interestingly, parameter space metrics such as the FIM are represented by classes prefixed with `PMat`. For instance, the natural gradient $\delta_{\text{nat}} = -\eta F^{-1}\nabla_{\mathbf{w}}\mathcal{L}$ applies the linear operator $\mathbf{w} \mapsto F^{-1}\mathbf{w}$ to the parameter space vector $\nabla_{\mathbf{w}}\mathcal{L}$, and can be implemented cleanly and concisely using `delta_nat = - eta * F.solve(nabla_L)`, even if it internally involves different operations for different layer types, and different approximation techniques.

Function space vectors `FVector` define objects associated to vectors of the output space, evaluated on a dataset of $n$ examples $X$. As an example, getting back to the linearization $\delta f(x, \mathbf{w}, \delta\mathbf{w}) = J(x, \mathbf{w})\delta\mathbf{w}$, we define $\delta\mathbf{f}(X) = (\delta f(x_1, \mathbf{w}, \delta\mathbf{w}), \ldots, \delta f(x_n, \mathbf{w}, \delta\mathbf{w}))$ as the $\mathbb{R}^{c \times n}$ function space vector of output changes for all examples of $X$. Gram matrices of the NTK are linear operators on this space, represented by objects prefixed with `FMat`. Borrowing from the vocabulary of differential geometry, we also define `PushForward` objects that are linear operator from parameter space to function space, and `PullBack` objects that are linear operator from function space to parameter space.

While the following consideration can be ignored upon first glance, the structure of the parameter space is internally encoded using a `LayerCollection` object. This gives the flexibility of defining our parameter space as parameters of a subset of layers, in order to treat different layers in different ways. An example use case is to use KFAC for linear layers parameters, and block-diagonal for GroupNorm layers, as KFAC is not defined for the latter.

### 2.2.2 CONCRETE REPRESENTATIONS

These abstract objects are implemented in memory using concrete representations. NNGeometry comes with a number of representations. Amongst them, most notably, are parameter space approximations proposed in recent literature (Ollivier, 2015; Martens & Grosse, 2015; Grosse & Martens, 2016; George et al., 2018), and an implicit representation for each abstract linear operator, that allows to compute linear algebra operations without ever computing or storing the matrix in memory.

`PMatDense` (resp `PMatDense`) and `PMatDiag` represent the full dense matrix and the diagonal matrix and need no further introduction. `PMatLowRank` only computes and stores $\mathbf{J}(X, \mathbf{w})$ the $c \times n \times d$ stacked Jacobian for all examples of the given dataset.

Next come representations that do not consider neural networks as black-box functions, but instead are adapted to the layered structure of the networks: `PMatBlockDiag` uses dense blocks of the FIM for parameters of the same layer, and puts zeros elsewhere, ignoring cross-layer covariance. `PMatQuasiDiag` (Ollivier, 2015) uses the full diagonal and adds to each bias element the interaction with the corresponding row of the weight matrix. `PMatKFAC` uses KFAC (Martens & Grosse, 2015) and its extension to convolution layers KFC (Grosse & Martens, 2016) to approximate each layer blocks with the kronecker product of 2 much smaller matrices, thus saving memory and compute compared to `PMatBlockDiag`. `PMatEKFAC` uses the EKFAC (George et al., 2018) extension of KFAC.

The last representation that comes with this first release of NNGeometry, `PMatImplicit`, allows to compute certain linear algebra operations using the full dense matrix, but without the need to ever store it in memory, which permits scaling to large networks (see experiments in section 3). As an illustration, the vector-matrix-vector product $\mathbf{v}^{\top}F\mathbf{v}$ can be computed using equation 3.

Each representation comes with its advantages and drawbacks, allowing to trade-off between memory and approximation accuracy. For a new project, we recommend starting with a small network using the `PMatDense` representation, then gradually switching to representations with a lower memory footprint while experimenting with actual modern networks.

While linear algebra operations associated to each representation internally involve very different mechanisms, NNGeometry's core contribution is to give easy access to these operations by using the same simple methods (figure 1).

### 2.2.3 GENERATORS

In order to compute FIMs and NTKs, we need to compute Jacobians $J(x, \mathbf{w})$ for examples $x$ coming from a dataset. NNGeometry's generator is the component that actually populates the representations by computing the required elements of the matrices, depending on the representation. While

a naive idea would be to loop through examples $x_i$, compute $f(x_i, \mathbf{w})$ and compute gradients with respect to parameters using PyTorch's automatic differentiation, it is rather inefficient as it does not make usage of parallelism in GPUs. NNGeometry's generator instead allows to use minibatches of examples by intercepting PyTorch's gradients and using techniques such as those in (Goodfellow, 2015) and (Rochette et al., 2019):

Let us consider $f(x, \mathbf{w}) : \mathcal{X} \times \mathbb{R}^d \to \mathbb{R}^c$. In order to simplify exposition, we focus on fully connected layers and suppose that $f$ can be written $f(x, \mathbf{w}) = \sigma_l \circ g_l(\cdot, \mathbf{w}) \circ \sigma_{l-1} \circ g_{l-1}(\cdot, \mathbf{w}) \circ \ldots \circ \sigma_1 \circ g_1(x, \mathbf{w})$ where $\sigma_k$ are activation functions and $g_k$ are parametric affine transformations that compute pre-activations $s_k$ of a layer using a weight matrix $W_l$ and a bias vector $b_k$ with the following expression: $s_k = g_k(a_{k-1}, \mathbf{w}) = W_k a_{k-1} + b_k$. For each example $x_i$ in a minibatch, we denote these intermediate quantities by superscripting $s_k^{(i)}$ and $a_k^{(i)}$. The back-propagation algorithm applied to computing gradients of a sum $S = \sum_i f(x_i, \mathbf{w})$ works by sequentially computing intermediate gradients $\frac{\partial f(x_i, \mathbf{w})}{\partial s_k^{(i)}}$ from top layers to bottom layers. Denote by $D\mathbf{s}_k = \left( \frac{\partial f(x_i, \mathbf{w})}{\partial s_k^{(1)}}^\top, \ldots, \frac{\partial f(x_i, \mathbf{w})}{\partial s_k^{(m)}}^\top \right)^\top$ the matrix obtained by stacking these gradients for a minibatch of size $m$, and $\mathbf{a}_k = \left( a_k^{(1)}, \ldots, a_k^{(m)} \right)$ the corresponding matrix of activations of the same layer. These are already computed when performing the backpropagation algorithm, then used to obtain the average gradient w.r.t the weight matrix by means of the matrix/matrix product $\frac{\partial}{\partial W_l} \left\{ \sum_i f(x_i, \mathbf{w}) \right\} = D\mathbf{s}_k^\top \mathbf{a}_k$. The observation of Goodfellow (2015) is that we can in addition obtain individual gradients $\frac{\partial f(x_i, \mathbf{w})}{\partial s_k^{(1)}}^\top a_k^{(i)\top}$, an operation that can be efficiently done simultaneously for all examples of the minibatch using the `bmm` PyTorch function.

While we used this already known trick as an example of how to make profit of minibatching, NNGeometry's generator incorporate similar tricks in several other places, including in implicit operations.

Instead of reimplementing backpropagation as is for example done by Dangel et al. (2019), we chose to use PyTorch's internal automatic differentiation mechanism, as it already handles most corner cases encountered by deep learning practitioners: we do not have to reimplement backward computations for every new layer, but instead we just have to compute individual gradients by intercepting gradients with respect to pre-activations $D\mathbf{s}_k$.

Other generators are to be added to NNGeometry in the future, either by using different ways of computing the Jacobians, or by populating representations using other matrices such as the Hessian matrix, or the KFRA approximation of the FIM (Botev et al., 2017).

## 3 Experimental showcase

Equipped with NNGeometry, we experiment with a large network: We train a 24M parameters Resnet50 network on TinyImagenet. We emphasize that given the size of the network, we would not have been able to compute operations involving the true $F$ without NNGeometry's `PMatImplicit` representation, since $F$ would require 2.3 petabytes of memory ($24M \times 24M \times 4$ bytes for float32).

### 3.1 Quality of FIM approximations

We start by comparing the accuracy of several `PMat` representations at computing various linear algebra operations. We use a Monte-Carlo estimate of the FIM, where we use 5 samples from $p(y|x)$ for each example $x$. Here, since this TinyImagenet is a classification task, $p(y|x)$ is a multinoulli distribution with the event probabilities given by the softmax layer. We compare the approximate value obtained for each representation, to a "true" value, obtained using the full matrix with the `PMatImplicit` representation. For trace and $\mathbf{v}^\top F \mathbf{v}$, we compare these quantities using the relative difference $\left| \frac{\text{approx} - \text{true}}{\text{true}} \right|$. For $F\mathbf{v}$, we report the cos-angle $\frac{1}{\|F\mathbf{v}\|_2 \|F_{\text{approx}} \mathbf{v}\|_2} \langle F\mathbf{v}, F_{\text{approx}} \mathbf{v} \rangle$, and for the solve operation, we report the cos-angle between $\mathbf{v}$ and $(F_{\text{approx}} + \lambda I)^{-1} (F + \lambda I) \mathbf{v}$.

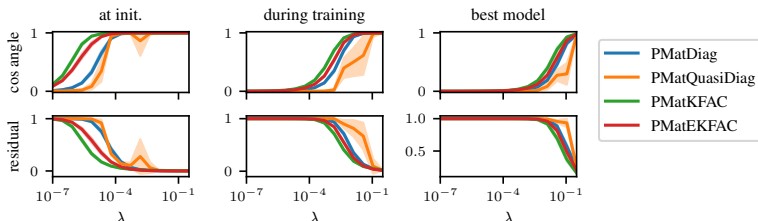

Figure 3: Residual $\frac{\|\mathbf{v}-\mathbf{v}'\|_2}{\|\mathbf{v}'\|_2}$ and cos angle between $\mathbf{v}$ and $\mathbf{v}' = (F_{\text{approx}} + \lambda I)^{-1}(F + \lambda I)\mathbf{v}$ for a 24M parameters Resnet50 at different points during training on TinyImagenet, using different approximations $F_{\text{approx}}$ of $F$, for $\mathbf{v}$ uniformly sampled on the unit sphere (higher is better).

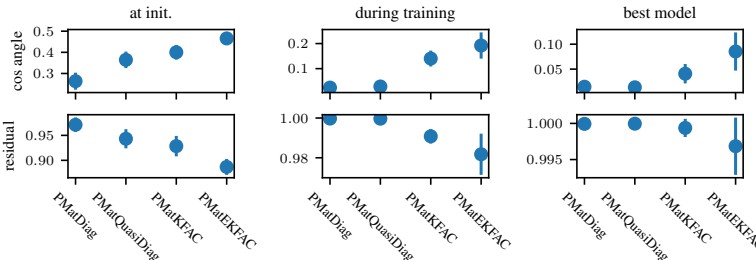

Figure 4: Cos angle between $F\mathbf{v}$ and $F_{\text{approx}}\mathbf{v}$ for a 24M parameters Resnet50 at different points during training on TinyImagenet, using different approximations $F_{\text{approx}}$ of $F$, for $\mathbf{v}$ uniformly sampled on the unit sphere (higher is better).

Since the latter is highly dependent on the Tikhonov regularization parameter $\lambda$, we plot the effect on the cos-angle of varying the value of $\lambda$. The results can be observed in figures 3, 4, 5, 6.

From this experiment, there is no best representation for all linear algebra operations. Instead, this analysis suggest to use `PMatKFAC` when possible for operations involving the inverse FIM, and `PMatEKFAC` for operations involving the (forward) FIM. Other representations are less accurate, but should not be discarded as they can offer other advantages, such as lower memory footprint, and faster operations.

## 3.2 Neural Tangent Kernel eigenvectors

In the line of Baratin et al. (2020); Paccolat et al. (2020), we observe the evolution of the NTK during training. We use the Resnet50 on the 200 classes of TinyImagenet, but in order to be able to plot a 2d matrix for analysis, we extract the function $f_{c_1,c_2}(x, \mathbf{w}) = (f(x, \mathbf{w}))_{c_2} - (f(x, \mathbf{w}))_{c_1}$, namely a binary classifier of class $c_2$ vs class $c_1$. We plot at different points during training i) the Gram matrix of examples from the 2 classes $c_1$ and $c_2$ (figure 7, top row) and ii) a kernel pca of points from classes $c_1$ and $c_2$ projected on the 2 first principal components (figure 7, bottom row). The Gram matrix is computed for valid set examples of classes $c_1$ and $c_2$.

On this larger network, we reproduce the conclusion of Baratin et al. (2020); Paccolat et al. (2020) that the NTK evolution is not purely random during training, but instead adapts to the task in a very specific way.

## 4 Conclusion

We introduced NNGeometry, a PyTorch library that allows to compute various linear algebra operations involving Fisher Information Matrices and Neural Tangent Kernels, using an efficient implementation that is versatile enough given current usages of these matrices, while being easy enough to save time for the user.

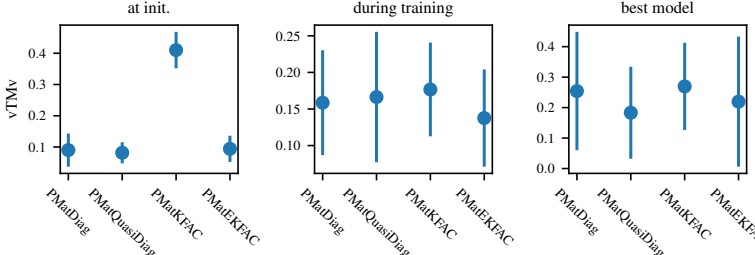

Figure 5: Relative difference between $\mathbf{v}^\top F \mathbf{v}$ and $\mathbf{v}^\top F_{\text{approx}} \mathbf{v}$ for a 24M parameters Resnet50 at different points during training on TinyImagenet, using different approximations $F_{\text{approx}}$ of $F$, for $\mathbf{v}$ uniformly sampled on the unit sphere (higher is better).

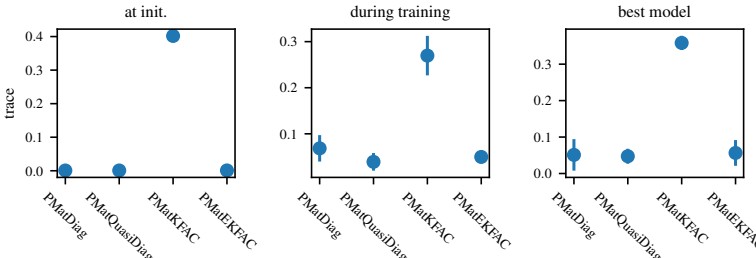

Figure 6: Relative difference of trace computed using $F_{\text{approx}}$ and $F$ (lower is better). As we observe, all 3 representations `PMatDiag`, `PMatQuasiDiag` and `PMatEKFAC` estimate the trace very accurately, since the only remaining fluctuation comes from Monte-Carlo sampling of the FIM. On the other hand, the estimation provided by `PMatKFAC` is less accurate.

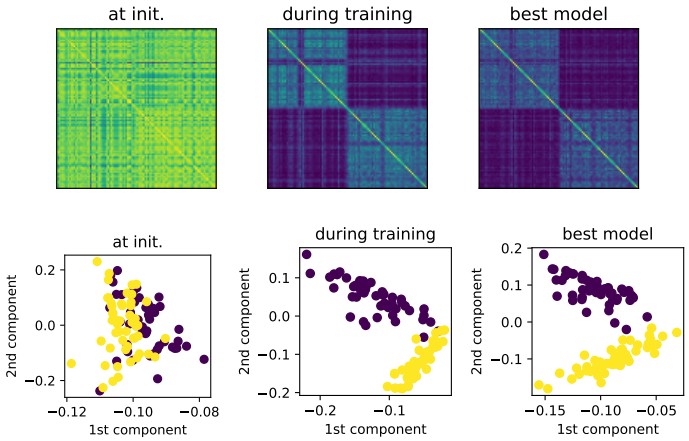

Figure 7: NTK analysis for 50 examples of class $c_1$ and 50 examples of class $c_2$ at various points during training. **(top row)** Gram matrix of the NTK. Each row and column is normalized by $\frac{1}{\sqrt{diag\,(G)}}$ for better visualization. We observe that the NTK encodes some information about the task later in training, since it highlights intra-class examples. **(bottom row)** Examples are projected on the 1st 2 principal components of the Gram Matrix at various points during training. While points are merely mixed at initialization, the NTK adapts to the task and becomes a good candidate for kernel PCA since examples become linearly separable as training progresses.

We hope that NNGeometry will help make progress across deep learning subfields as FIMs and NTKs are used in a range of applications.

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

## A    APPENDIX

You may include other additional sections here.

