# OpenReview forum: "NNGeometry: Easy and Fast Fisher Information Matrices and Neural Tangent Kernels in PyTorch"
_ICLR.cc/2021/Conference — Reject_

### Official Review · AnonReviewer1 · 2020-10-19
**The paper described the details of an efficient PyTorch implementation of Fisher information matrix and neutral tangent kernel. To achieve this, the author adopted several approximation and computational tricks from previous work in the literature.**

**Rating:** 5
**Confidence:** 2

**Review:**

The paper provides a PyTorch implementation for applying Fisher information matrix and neutral tangent kernel as a linear operator. This efficient implementation relies on several existing approximation methods and computational tricks. I think the work is practical as it can be used for many applications, some of which from works that are cited in this paper. The paper is reasonably well-written, but some details are missing. I'm mostly concerned with the novelty and and significance of this work, although I need to admit I'm unfamiliar with the acceptance criteria for papers on software implementation.

1. I think most of the techniques in the paper are from previous work. Thus I'm not sure there's enough novelty for an acceptance. The paper did not highlight any improvement in scaling. I believe all the speedup comes from taking advantage of GPU operation using the trick from Goodfellow, and the collection of matrix approximation methods that the paper cited.

2. The paper focused on implementation, but surprisingly the experiments did not include any scaling/runtime results. I think the authors should show the efficiency gain, and demonstrate the type the applications that are enabled by it.

---

> ### Author Response · Authors · 2020-11-16
> **"Implementation issues" topic in call for papers**
>
> Thanks for your review. As we also answered to AnonReviewer3, the main contribution is the high-level API for manipulating FIMs and NTKs, that we thought could fit in the “implementation issues” topic of ICLR’s call for paper. We do not claim novelty in how things are implemented at low-level.
>
> Even if it is limited, we would like to emphasize that our experiments could not be performed without NNGeometry, since it involved computing matrix-vector products Fv for 24M x 24M matrix F, a computation that is performed implicitely in NNGeometry.

---

### Official Review · AnonReviewer3 · 2020-10-19
**A new PyTorch package for computing the FIM and NTK is really nice, but it's unclear if it's a research contribution.**

**Rating:** 4
**Confidence:** 3

**Review:**

This paper describes a new PyTorch package, NNGeometry, for computing complicated neural network objects, such as the Fisher Information Matrix (FIM) and the Neural Tangent Kernel (NTK). The package uses an abstract representation to allow the user to implicitly choose between different approximations to these objects and automatically makes a bunch of efficient choices "under the hood" for the user. The paper concludes with a selection of experiments that compare these approximations as well as verify their validity against Monte-Carlo estimates of these matrices.

The package seems like a really fantastic contribution to PyTorch. I will definitely try it out and most likely use it for my research.

On the other hand, I am worried that this is more of an engineering contribution than a research paper. To the point, all of the methods described in this paper and implemented in the package are based on methods present in other papers. Furthermore, the authors do not use the package to make any novel research contributions. Instead, they first show how their FIM approximations compare to Monte-Carlo estimates, and then they use their NTK approximations to reproduce some NTK results after training.

I feel pretty undecided about whether ICLR should publish this paper. On the one hand, I think the package is very nice; it accumulates a bunch of useful approximations and allows for an efficient interface for switching between them. However, I don't know how significant this is as a research contribution; the authors propose no new methods nor do they achieve any novel results with their package. For now I think that the paper is not quite able to be published at ICLR, but I'm interested to hear what the authors and other reviewers have to say.

To that point, I wonder if the authors might use their package to show how it allows researchers to perform certain types of experiments that couldn't have previously been performed.

Some minor comments and a question:
- I think on Figure 3, the x-axis is lambda? It would be great if a label was included.
- In the References section, Paccolat et al. (2020) is missing its arxiv number.
- Could the authors include some detail about the type of Monte-Carlo experiments they did to compare for Figures 3-8?

---

> ### Author Response · Authors · 2020-11-16
> **"Implementation issues" topic in call for papers**
>
> Thanks for kind comments and suggestions. The main contribution is the high-level API for manipulating FIMs and NTKs, that we thought could fit in the “implementation issues” topic of ICLR’s call for paper, as we think the lack of such a library hampers research progress.
>
> In our revision, we took your comments into account:
>  - the missing 'lambda' was due to a problem with embedded fonts in pdf, that should be solved with this new upload
>  - we fixed the incorrect reference
>  - we added a description of Monte-Carlo estimation in the experiment section. In short, this is for estimating p(y|x) for classification where p(y|x) is a multinoulli distribution with event probabilities given by the softmax of the output.
>
> To answer your comment about experiments that couldn't have previously been performed, we can take the example of the EWC paper. In their derivation, they use the FIM, but in their experiments, they only take the diagonal of the FIM. But there is no particular reason for F to be diagonally dominant, so we do not really know if F is well approximated by its diagonal. In that regards, we do not know if the success of EWC really comes from FIM information. Using NNGeometry (that did not exist at that time), they could have tried their method using more accurate approximations on small tasks, then scaled to larger network using less memory intensive approximations, by using the exact same code.

---

### Official Review · AnonReviewer2 · 2020-10-28
**A PyTorch library for working with Fisher Information Matrices and Neural Tangent Kernels**

**Rating:** 7
**Confidence:** 3

**Review:**

### Summary

The Fisher information matrix and the neural tangent kernel matrix have been used in several recent papers to provide insight into deep neural networks, but operations involving these matrices have so far been less well supported in frameworks such as Tensorflow and PyTorch.  The current paper describes a new library (NNGeometry) for working with these matrices in PyTorch, incorporating approximate representations from prior work that are appropriate to matrix-vector product computations and evaluation of quadratic forms.

The main contribution of the paper is the software library; no new analysis or algorithms are presented.  However, this seems like a useful software artifact for the particular problem that it tackles.  I recommend accepting it if there is space.

### Additional details

- The cosine angle measure is a somewhat peculiar choice for comparing approximate matrix-vector products and solves with their reference versions.  It is not only the direction of these vectors that matters, but also their magnitude, and so a normwise relative error seems much more natural.  For the (regularized) linear system, the natural choice might be a normwise relative residual.
- While I understand the space constraints, I would have liked to see more discussion of the concrete matrix representations (Section 2.2.2).  In particular, there is not even a one-sentence description of the KFAC or EKFAC representation; only the name and a reference are given.  Adding a sentence or two would probably not break the page limit, and would make the paper more accessible to readers unfamiliar with those prior papers.

### Minor points

- Throughout, the authors should carefully check choice of citation style and how the citation reads (e.g. "efficient techniques such as Goodfellow (2015)" seems to refer to the cited paper as a technique; I suggest rewording to "... such as those in Goodfellow (2015)" (or "of Goodfellow (2015)").
- Typos: "inn" to "in" (top of page 5), "with on a large network" - "with a large network" (start of Section 3)

### Update

Thanks to the authors for their responses, particularly to the other reviewers who were more critical.  I still believe this is worth publishing if there is room, with the caveat stated before: this is an implementation paper, and does not introduce new algorithms or analysis.  If it is accepted, it should be accepted on this basis.

---

> ### Author Response · Authors · 2020-11-16
> **Normwise relative error and KFAC**
>
> Thanks for good suggestions, and a good rating!
>
> We revised our paper to include your suggestions:
>  - we added relative norm to our plots involving comparison of Fv for different approximations of F (figure 3 and 4).
>  - we added a very quick description of KFAC and EKFAC. We however chose not to go too much into detail, as it would overload the paper in maths.
>
> We also fixed citation style and other typos.
>
> Thanks again for your very valuable feedback.

---

### Official Review · AnonReviewer4 · 2020-10-28
**PyTorch library for easy/fast computation of Fisher matrices; many parts unclear and possibly incorrect**

**Rating:** 4
**Confidence:** 4

**Review:**

Summary: This paper introduces a new PyTorch library for computing Fisher Information Matrices and Neural Tangent Kernel (NTK) in deep learning; with applications ranging from Frobenius norm regularization, second-order optimization, and generalization analysis. The authors begin by providing a background on Fisher matrices and NTK and then present the main components of their proposed NNGeometry library (consisting of Layer Collection, Generator, and  Concrete Representations modules). A brief experimental study is provided in the last part of the paper.

Assessment: While I think that a clean and effective computational library for implementation of Fisher matrices would greatly benefit the DL/ML research community, the current work falls short of the standards of an ICLR publication in numerous ways.

Detailed Comments:
- Towards the end of the introduction, the authors claim "NNGeometry aims at making use of these appoximations effortless...". I really do not see how NNGeometry is capable of doing this. One of the reasons why natural gradient methods (and its approximations such as K-FAC) are rather difficult to implement is that they are "white-box" optimization methods and very much model-dependent, i.e., different model architectures means that the Fisher matrices (and their approximations) can look very different from each other. Please see [1], [2] for the K-FAC approximations needed for convolutional and recurrent networks respectively. As an aside, I do not believe there is even a good open-source code for K-FAC on RNNs especially given the complexity involved. I did not find anywhere in the paper how NNGeometry addresses approximations for different types of layers.

- A lot of spelling mistakes and strange notations throughout the paper. Here are a few I found- "jacobian" is not capitalized throughout, the big "O" notation in Equations 3 and elsewhere in the paper are all small "o"'s

- The discussion in Section 1.2 is a bit wordy and not precise mathematically at times. I would suggest cutting down and citing [3]. Also, somewhere in the introduction, I think the authors should make a note of the distinction between the empirical Fisher matrix and the true Fisher matrix (and perhaps cite [4]); and be clear about which one they are working with.

- Given that this is a library concerned with computing FIM/NTK; there should be some comparisons with existing open-source libraries such as JAX and Neural Tangents?

- Lots of issues in Sections 2.2.1 and 2.2.2. It is not exactly clear where the authors are trying to do here; and there are many imprecise/incorrect mathematical statements throughout. I believe that the purpose of Section 2.2.1 was to describe that the FIM defines a Riemannian metric on the parameter space; and that the FIM is a representation of this metric in coordinate form. This is certainly true- but I cannot see the connection of this to the NNGeometry framework. Another purpose of this subsection was the notion of duality; for example, which objects may be pushed forward/pulled back (to be more precise, which ones live on the tangent and cotangent spaces). I would encourage the authors to look at the publicly-available JAX documentation/tutorial; where it is explained nicely how all of the theory + code fits together; JVP (Jacobian-vector products) / forward-mode autodiff <--> pushforward map of tangent spaces, VJP (vector-Jacobian products) / reverse-mode autodiff <--> pullback of cotangent spaces.

- It would be great if the authors were more clear and explained explicitly the tricks in the sentence "NNGeometry's generator incorporate similar tricks in several other places, including in implicit operations". Many of these types of tricks are known to practitioners who have had to implement FIM (and its approximations); so I am curious what is the novelty provided by NNGeometry's generator here.

References:

[1] Grosse, Roger, and James Martens. "A kronecker-factored approximate fisher matrix for convolution layers." International Conference on Machine Learning. 2016.

[2] Martens, James, Jimmy Ba, and Matt Johnson. "Kronecker-factored curvature approximations for recurrent neural networks." International Conference on Learning Representations. 2018.

[3] Martens, James. "New insights and perspectives on the natural gradient method." arXiv preprint arXiv:1412.1193 (2014).

[4] Kunstner, Frederik, Philipp Hennig, and Lukas Balles. "Limitations of the empirical Fisher approximation for natural gradient descent." Advances in Neural Information Processing Systems. 2019.

---

> ### Author Response · Authors · 2020-11-16
> **Clarifications**
>
> Thanks for a thorough review with very relevant suggestions. We would like to first restate clearly (sorry if it was not made explicit enough in the paper) that the main contribution of NNGeometry is to offer a high level API for manipulating FIMs and NTKs, we do not claim novelty in how things are implemented at lower level.
>
> In this new revision, we modified section 1.2 and 2.2.1 to address your comments, and we hope the following will clarify some other points:
>
>  - "Towards the end of the introduction, the authors claim "NNGeometry aims at making use of these appoximations effortless...". I really do not see how NNGeometry is capable of doing this. [...]"
>
> => This is precisely the “white-box” problem that is addressed by NNGeometry: With K-FAC for instance, it means that you can implement a natural gradient optimizer by writing natgrad = F_kfac.solve(grad,epsilon) regardless of the layer types, where grad and natgrad are PVectors, i.e. they contain all parameters of a network. Internally, NNGeometry will automatically choose between K-FAC for fully connected layers, and KFC for convolutional layers. While we do not currently support RNNs, here is a list of nn.Module layers in PyTorch that are already implemented: nn.Linear, nn.Conv2d, nn.GroupNorm, nn.BatchNorm2d (eval mode only). This already covers a range of modern deep architectures, and other layers will be implemented in the near future.
>
>  - "[...] "jacobian" is not capitalized throughout, the big "O" notation in Equations 3 and elsewhere in the paper are all small "o"'s"
>
> => Thanks for the typo with “Jacobian”! However, little-o is a different symbol from big-O, so it is on purpose that it is denoted with a small “o”. Please see https://en.wikipedia.org/wiki/Big_O_notation#Little-o_notation for differences, and https://en.wikipedia.org/wiki/Taylor's_theorem#Statement_of_the_theorem for Taylor expansion using little-o.
>
>  - "The discussion in Section 1.2 is a bit wordy and not precise mathematically at times. I would suggest cutting down and citing [3]. Also, somewhere in the introduction, I think the authors should make a note of the distinction between the empirical Fisher matrix and the true Fisher matrix (and perhaps cite [4]); and be clear about which one they are working with."
>
> => Thanks for suggesting reference [3], we added it to section 1.2. In the paper we try to stay as general and concise as possible, this is why there is no mention of the discussion about empirical vs true Fisher. In section 2.1 we describe the true Fisher, and in the experiments section we use a Monte-Carlo estimate of the true Fisher as stated in section 3.1 . Using NNGeometry, you can also work with the 2nd moment of the gradients (a.k.a. the empirical Fisher), and we added a comment about it at the end of section 2.1. But as suggested by your reference [4] and other authors (e.g. Thomas et al. AISTATS 2020), there is some confusion about it, so we preferred to refrain from putting too much emphasis on this ongoing debate.
>
>  - "Given that this is a library concerned with computing FIM/NTK; there should be some comparisons with existing open-source libraries such as JAX and Neural Tangents?"
>
> => JAX and Neural Tangents are designed for different purposes than NNGeometry, even if some features overlap. For instance the experiments of section 3.1 “Quality of FIM approximations” cannot be implemented with JAX without having to implement each operation for each approximation method (e.g. what NNGeometry does for PyTorch). In that regard, there is no direct competitor to NNGeometry.
>
>  - "Lots of issues in Sections 2.2.1 and 2.2.2. [...]"
>
> => The purpose of section 2.2.1 is only to relate mathematical objects defined in previous sections, to Python objects defined by NNGeometry. In the revision, we modified this section to make it more clear. However,  we think that a discussion about the precise nature of parameter, function space, and linear maps between the tangent space and the cotangent space is out of the scope of the library for most users, e.g. for a continual learning practitioner there is no notion of Riemannian geometry involved in EWC.
>
> - "It would be great if the authors were more clear and explained explicitly the tricks [...]"
>
> => The novelty of NNGeometry does not reside in its generator, but rather in the high level API to manipulate FIMs and NTKs. The purpose of this section was to make clear that NNGeometry’s generator does not naively loop through examples in order to compute per-example gradients, but we instead have an implementation that is efficient enough to perform experiments on modern networks such as demonstrated in the experiments section. We however do not claim novelty for low-level implementations.
>
> Once again, thanks for very valuable feedback. We hope to have answered your concerns.
>
> reference:
> Thomas, Valentin, et al. "On the interplay between noise and curvature and its effect on optimization and generalization." AISTATS, 2020.

---

> > ### Author Response · Authors · 2020-11-20
> > **Did we answer your comments?**
> >
> > Thanks again for your time.
> >
> > The first phase of response period is going to end soon. We would like know if our answers and our revised draft addressed your initial comments?  If you have other concerns, we will be happy to address them before the rebuttal period is over.

---

### Decision · Program_Chairs · 2021-01-07
**Final Decision**

**Decision:**

Reject

**Comment:**

This paper provides a high-level API for working with Neural Tangent Kernels (NTK) and Fisher Information Matrices (FIM). This is an implementation paper, but such concepts are clearly useful in many tasks. However, such methods are available in many in-house code (almost every paper on FIM / NTK uses such methods) I would not say it hampers the progress.


Pros: - The proposed methods are already widely used by the communities in the in-house codes, but no single library is available
          - The library implements state-of-the art approaches

Cons: - This is an implementation library, and no benchmarking is available. More testing is needed to showcase the library.
           - Matrix-by-vector products are provided, but typical operations include many more, like Lanczos method for approximating solutions of linear systems and matrix function by vector product, or randomized SVD to compute low-rank approximations. I believe that this should be the part of the library in order to make it a serious competitor for existing "local implementations". Also, such kind of methods would be later or sooner part of big libraries such as PyTorch and Tensorflow.